# A modelled analysis of the impact of COVID-19-related disruptions to HPV vaccination

Louiza S Velentzis[1,2]*[†], Megan A Smith[1]*[†], James Killen[1], Julia ML Brotherton[2,3], Rebecca Guy[4], Karen Canfell[1]

[1]Daffodil Centre, The University of Sydney, a joint venture with Cancer Council NSW, Sydney, Australia; [2]School of Population and Global Health, University of Melbourne, Victoria, Australia; [3]Australian Centre for Prevention of Cervical Cancer, Victoria, Australia; [4]The Kirby Institute, University of New South Wales, Sydney, Australia

**Abstract** COVID-19 disrupted school attendance in many countries, delaying routine adolescent vaccination against human papillomavirus (HPV) in some settings. We used *Policy1-Cervix*, a dynamic model simulating HPV transmission, natural history, vaccination, cervical screening, and diagnosis of HPV-related cancers, to estimate the impact on HPV-related cancers from disruptions to HPV vaccination in a high-income setting. A baseline scenario of no disruption to HPV vaccination was modelled, which assumed uptake of the nonavalent vaccine at the age of 12 by 82.4% of females and 75.5% of males, as is the coverage in Australia. Additional lifetime HPV-related cancer cases were calculated for three disruption scenarios affecting one birth cohort (2008; aged 12 in 2020) compared to the baseline scenario: (1) 1-year delay (no doses missed); (2) 1- to 7-year delay (slow catch-up); (3) no catch-up (herd effects only). A fourth scenario assumed no catch-up HPV vaccination for two birth cohorts, that is all individuals born in 2008 and in 2009 missed vaccination (worst-case scenario). Compared to 1532 HPV-related cancer cases estimated for the baseline no disruption scenario, we found a 1-year delay could result in ≤0.3% more HPV-related cancers (*n* = 4) but the increase would be greater if catch-up was slower (5%; *n* = 70), and especially if there was no catch-up (49%; *n* = 750). Additional cancers for a single missed cohort were most commonly cervical (23% of the additional cases) and anal cancers (16%) in females and oropharyngeal cancers in males (20%). In the worst-case scenario of two birth cohorts missing vaccination, ≤62% more HPV-related cancers would be diagnosed (*n* = 1892). In conclusion, providing catch-up of missed HPV vaccines is conducted, short-term delays in vaccinating adolescents are unlikely to have substantial long-term effects on cancer.

*For correspondence:
louizav@nswcc.org.au (LSV);
megan.smith@nswcc.org.au
(MAS)

[†]These authors contributed equally to this work

## Editor's evaluation

The study presents important findings for public health authorities and policymakers to enable them to make evidence-based decisions when deciding on how to manage the effect of HPV vaccination disruptions. This study is particularly relevant in light of the efforts of the WHO to achieve global elimination of cervical cancers. The findings are convincing and the model used is appropriate.

## Introduction

Australia was the first country to implement a fully government funded National HPV Vaccination program. Human papillomavirus (HPV) vaccination has been routinely offered through schools, to girls aged 12–13 years since 2007 and to boys aged 12–13 years since 2013. Additionally, catch-up

HPV vaccination was offered to females aged 14–26 years in 2007–2009 and to males aged 14–15 years in 2013–2014. In 2018, a three-dose schedule with the quadrivalent HPV vaccine which protects against four HPV types (6, 11, 16, and 18) was replaced by a two-dose schedule with the nonavalent HPV vaccine (HPV9), protecting against an additional five oncogenic HPV types (31, 33, 45, 52, and 58). Since 2017, free catch-up of vaccinations under the Australian National Immunization Program, including the HPV vaccine, is available to all people aged under 20 years and is administered in primary care (*Australian Government Department of Health and Aged Care, 2022*). HPV course completion rates by age 15 were 79.8% among girls and 77.0% among boys who reached the age of 15 years in 2019 (*National Centre for Immunisation Research and Surveillance Australia, 2020*).

Following the World Health Organization's declaration of the COVID-19 pandemic in March 2020, strict national- and state-level infection control measures were introduced in Australia. Social and physical restrictions affected essential health services including the delivery of school-based vaccination programmes due to school closures from imposed lockdowns in 2020 and in some states in 2021 (*National Centre for Immunisation Research and Surveillance Australia, 2021a*; *Parliament of Victoria, 2021*). Globally, the World Health Organization reported the largest continued decline in overall vaccinations in the last three decades and a loss of over a quarter of the HPV vaccine coverage achieved in 2019 (*World Health Organization, 2022*).

Infection with oncogenic HPV types is a major causal factor for the development of cervical cancer and for a fraction of cancers of the anus, oropharynx, penis, vagina, and vulva (*IARC Working Group on the Evaluation of Carcinogenic Risks to Humans, 2012*). Except for cervical screening, organized screening is not generally available for any of the other HPV-related cancers. A previous analysis of the burden of HPV-related disease in Australia estimated that 80% of the estimated 1544 HPV-associated cancers in 2012 were attributable to types preventable by the quadrivalent vaccine (HPV16/18), with an additional 9% attributable to the five additional types prevented by the nonavalent vaccine (*Patel et al., 2018*).

The aim of this study was to estimate the additional lifetime HPV-related cancer cases in men and women that could be caused by HPV vaccination delays or missed doses due to the pandemic by using a simulation model to project the long-term outcomes from different potential disruption scenarios, using Australia as an example. The study was designed prior to the emergence of actual 2020–2021 vaccination disruption data, and hence, a number of potential disruption scenarios were considered to encompass a wide range of possibilities.

**Table 1.** Estimated number of human papillomavirus (HPV)-related cancer cases and cases prevented for vaccinated and unvaccinated cohorts, and four vaccination disruption scenarios.

| Modelled scenarios | Outcomes in 2008 birth cohort[*] | | | | | Outcomes in 2008 and 2009 birth cohorts[†] | | | | |
| --- | --- | --- | --- | --- | --- | --- | --- | --- | --- | --- |
| | Total cases | Prevented cases (unvax comparator) | Additional cases (compared to no disruption) | % Prevented (unvax comparator) | % Additional (compared to no disruption) | Total cases | Prevented cases (unvax comparator) | Additional cases (compared to no disruption) | % Prevented (unvax comparator) | % Additional (compared to no disruption) |
| Unvaxed | 3923 | | | | | 7847 | | | | |
| No disruption | 1532 | 2583 | | 63 | | 3061 | 4876 | | 63 | |
| S1: 1-year delay | 1537 | 2579 | 4 | 63 | 0.3 | 3066 | 4781 | 5[‡] | 63 | 0.2 |
| S2: 1- to 7-year delay | 1603 | 2513 | 70 | 61 | 5 | 3133 | 4714 | 72[‡] | 62 | 2 |
| S3: no catch-up | 2282 | 1833 | 750 | 45 | 49 | 3846 | 4001 | 785[‡] | 53 | 26 |
| S4: no catch-up (two missed cohorts) | 2503 | 1613 | 970 | 39 | 63 | 4954 | 2893 | 1892 | 40 | 62 |

No disruption: uninterrupted HPV vaccination in females and males at age 12 with status quo uptake; scenario 1: disruption with rapid catch-up, 1-year delay in HPV vaccine catch-up; scenario 2: disruption with slow catch-up, 1- to 7-year delay in HPV vaccine catch-up; scenario 3: disruption with no HPV vaccine catch-up (herd effects only; 2008 cohort affected); scenario 4: disruption with no HPV vaccine catch-up (herd effects only; 2008 and 2009 cohorts affected).

[*]Includes outcomes specifically for the cohort consisting of females and males born in 2008 (any effects on the 2009 cohort are not included).

[†]Includes outcomes specifically for the cohort consisting of females and males born in either 2008 or 2009.

[‡]Differences between these additional cases compared to additional cases in the outcomes specific to the 2008 cohort (left-hand side of table) are additional cases in unvaccinated individuals in the 2009 cohort, due to a loss in the indirect protection they received from vaccination of the 2008 cohort in the *No disruption* scenario due to herd effects.

vax: vaccine. unvaxed: unvaccinated i.e. assuming no HPV vaccination in cohort(s).

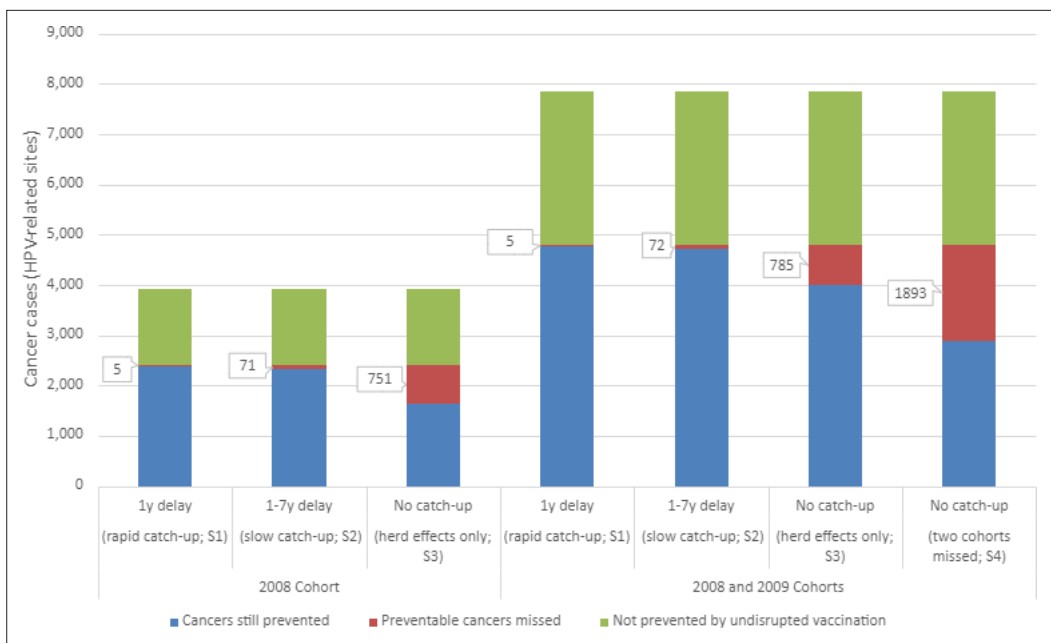

**Figure 1.** Estimated lifetime human papillomavirus (HPV)-related cancer cases from four modelled scenarios. Scenarios include two HPV vaccination catch-up scenarios [S1: 1-year delay in vaccination (rapid); S2: 1- to 7-year delay in vaccination (slow)] and two scenarios modelling the absence of vaccination catch-up, varying in the cohort affected (S3: scenario 3 affecting the 2008 birth cohort; S4: scenario 4 affectiong the 2008 and 2009 birth cohorts).

## Results

In the baseline scenario of no interruption to HPV vaccination, we estimate 2583 HPV-related cancer cases would be prevented in the 2008 cohort due to HPV vaccination, equating to 63% of all HPV-related cases (*Table 1*). The first disruption scenario (1-year delay) would result in four (0.3%) additional lifetime cancer cases and slightly fewer cases prevented overall (2579). Under a slow vaccination

**Table 2.** Estimated number of cancer cases in modelled scenarios according to sex and cancer type.

| Modelled scenarios | Total cases (additional compared to no disruption) | Females, *N* (additional compared to no disruption) | | | | | Males, *N* (additional compared to no disruption) | | |
|---|---|---|---|---|---|---|---|---|---|
| | | Anal | Cervical | Oropharyngeal | Vaginal | Vulvar | Anal | Oropharyngeal | Penile |
| 2008 cohort | | | | | | | | | |
| Unvaxed | 3923 | 489 | 788 | 185 | 162 | 729 | 389 | 911 | 271 |
| No disruption | 1532 | 100 | 62 | 86 | 61 | 580 | 74 | 423 | 146 |
| Scenario 1 | 1537 (4)* | 101 (1) | 63 (1) | 86 | 61 | 580 | 75 (1) | 424 (1) | 146 |
| Scenario 2 | 1603 (70)* | 114 (14) | 74 (12) | 90 (4) | 64 (3) | 586 (6) | 85 (11) | 440 (17) | 150 (4) |
| Scenario 3 | 2282 (750)* | 236 (136) | 250 (188) | 121 (35) | 96 (35) | 632 (52) | 175 (101) | 593 (170) | 180 (34) |
| 2008 and 2009 cohorts | | | | | | | | | |
| Unvaxed | 7847 | 978 | 1576 | 370 | 324 | 1458 | 778 | 1822 | 542 |
| No disruption | 3061 | 199 | 125 | 172 | 122 | 1160 | 148 | 845 | 292 |
| Scenario 4 2008 cohort | 2503 (971)* | 276 (176) | 303 (241) | 131 (45) | 107 (46) | 647 (67) | 205 (131) | 644 (221) | 190 (44) |
| 2009 cohort | 2451 (922) | 268 (169) | 285 (222) | 129 (43) | 104 (43) | 644 (64) | 199 (125) | 634 (212) | 188 (42) |
| Scenario 4 total | 4954 (1892)* | 544 (345) | 588 (463) | 260 (88) | 211 (89) | 1291 (131) | 404 (256) | 1278 (433) | 378 (86) |

Sum of cases may not add to 'total cases' due to rounding.

Unvaxed: assuming no HPV vaccination in cohort(s); No disruption: HPV vaccination in females and males at age 12 with coverage of 82.4% in females and 75.5% in males; scenario 1: disruption with rapid catch-up, 1-year delay; scenario 2: disruption with slow catch-up, 1- to 7-year delay; scenario 3: disruption with no catch-up (herd effects only; 2008 cohort affected); scenario 4: disruption with no catch-up (herd effects only; 2008 and 2009 cohorts affected).
*The number of additional cases presented in the table does not always match the difference between case figures stated for individual scenarios in *Table 2*, due to rounding.

catch-up strategy or in the worse cases – no vaccination catch-up for either one or two cohorts – an additional 70 (5%), 750 (49%), and 1892 (62%) cancer cases, respectively, were predicted to occur. *Figure 1* presents the estimated lifetime HPV-related cancer cases for all scenarios for the 2008 cohort (left) and combined 2008 and 2009 cohorts (right).

*Table 2* shows a breakdown of HPV-related cancer cases developed over a lifetime by cancer type and sex for the disruption scenarios. In the worst-case scenarios of missed HPV vaccination, most of the additional cancers occurred in females, and the additional cancers were most commonly cervical and anal cancers (females) and oropharyngeal, followed by anal cancers (males). When combined across both sexes, the additional cancers were most commonly oropharyngeal (>80% of which were in males) and anal cancers (mostly in females).

In the main analysis, we explicitly modelled primary HPV cervical screening as also occurring in these cohorts, which would allow the cervical screening programme to compensate, to some extent, for cervical cancers not prevented via HPV vaccination. In the sensitivity analysis that used an incidence-based approach, the number of additional cervical cancer cases predicted was larger and cervical cancers would have comprised around 35% of the additional HPV-related cancer cases (rather than around 17–25%, when screening was modelled) (*Supplementary file 1D*). The percentage increase in cases was similar but somewhat higher (e.g. 54% increase in cases for one missed cohort, compared to 49% increase when screening was modelled).

## Discussion

This is the first study to estimate the long-term potential impact of missed HPV vaccinations across HPV-induced cancers while accounting for natural history, sexual behaviour, and herd immunity. We found that rapid catch-up of HPV vaccination in young adolescents within a year meant that even extreme disruptions where all vaccinations were delayed had a minimal long-term effect on cancer cases. A slower pace of recovery, however, or, in the worst case, missed vaccination in one or two cohorts was associated with a higher excess burden of cancers. While herd effects from vaccination still prevented close to half of HPV-related cancers, missing a single cohort entirely would result in an estimated 49% more HPV-related cancers in that cohort compared to no disruption.

In Australia, initial national figures show for adolescents aged 11 to <15 years who received their first dose in 2020, 74.7% of girls and 72.6% of boys received their second dose in the same calendar year, compared to 86.2% and 84.3%, for girls and boys, respectively, in 2019 (*National Centre for Immunisation Research and Surveillance Australia, 2020*). The difference in numbers suggest vaccination delays in second doses, possibly due to COVID-19 effects including school closures during lockdowns, and decreased school attendance due to infections. These data do not provide insight into the impact on first doses. In line with WHO recommendations, final coverage data are reported as those who are vaccinated by age 15, and this information is not yet available for the most affected cohorts (aged 12–13 in 2020 and 2021; born approximately 2007–2009; turning 15 in 2022–2024) (*National Centre for Immunisation Research and Surveillance Australia, 2021b*). Reports from other countries indicate that COVID-19 has delayed progress in achieving high coverage of HPV vaccination. For example, a US study found recovery after a marked decline in HPV dose administration in March to May 2020 (median 64–71%) was not fully achieved during June to September 2020 (*Patel Murthy et al., 2021*).

In November 2020, WHO released a global strategy calling for all countries to take action to achieve the elimination of cervical cancer as a public health problem within the next century (*World Health Organization, 2020*). In the near future most lives will be saved globally through increased uptake of cervical screening and access to cancer treatment (*Canfell et al., 2020*), but in the longer-term maintaining high coverage of HPV vaccination will be critical in reducing mortality and achieving global elimination (*Canfell et al., 2020*; *Brisson et al., 2020*). Globally, the timeframe to reach elimination is likely to be impacted by pandemic-induced delays in HPV vaccination delivery and also by global shortages of the HPV vaccine estimated to last until 2024 thus delaying the introduction of vaccination in low- and middle-income countries (*World Health Organization (WHO), 2018*). In countries predicted to achieve cervical cancer elimination in the relatively short term, such as Australia, short delays of 1 year are not predicted to delay elimination (*Smith et al., 2021*). There are signs, however, of a decrease in vaccine confidence in Europe around the time of the COVID-19 pandemic,

with lower confidence for the HPV vaccine reported in 17 countries in 2022 compared to 2020 (*de Figueiredo et al., 2022*).

The strengths of this study include evaluating the impact of the COVID-19 pandemic across different HPV-related cancers, in both women and men and across a wide range of possibilities for vaccination disruption – two of which assumed a delay in HPV vaccination and two scenarios where the affected birth cohorts remained unvaccinated. Some limitations should also be noted. Even in the context of some cohorts missing vaccination entirely, herd effects still prevented a considerable proportion of HPV-related cancers overall; these effects would be smaller in settings with lower coverage (*Brisson et al., 2016*). This is unlikely to affect our finding that catch-up within a year minimizes the impact of disruptions, and so would likely strengthen the case for rapid catch-up in settings where herd effects are likely to be lower. The model also assumed that cervical screening would continue in birth cohorts offered HPV9, and, based on our sensitivity analysis that removed explicit modelling of screening, this prevented some cancers that would otherwise have occurred due to missed vaccination. In practice, screening may not continue in the same way in cohorts offered HPV9 as it would be much less cost-effective (*Simms et al., 2016*). As yet, no policy decision has been made for screening in these cohorts, however, and screening may in future be tailored based on individual vaccination status, rather than age eligibility for vaccination. The model did not incorporate any potential changes in sexual behaviour which may affect estimates.

Australia is a relatively high HPV vaccination coverage setting. Outcomes may be less favourable in a lower coverage setting, as there would be less protection from herd effects; however, the impact of disruptions might also be smaller in a setting with lower coverage, since a lower coverage programme would be less effective. Nevertheless, the finding that if catch-up is performed expeditiously then it mitigates much of the effect from vaccination delays, is likely to hold in other settings. In a previous study modelling the health impacts of HPV vaccination hesitancy in Japan from 2013 to 2019 and the potential effects of restoring coverage to 70% with catch-up vaccination in 2020 is informative as it demonstrates that multi-age HPV catch-up vaccination, such as after catastrophic falls in coverage in Japan, would be effective in mitigating the effects (*Simms et al., 2020*).

Our findings provide evidence that rapid catch-up of missed HPV vaccine doses would minimize the long-term impact of vaccination programme disruptions, but delays would result in preventable HPV-related cancer cases in future.

## Methods
### Model
We used *Policy1-Cervix*, an established modelling platform of dynamic HPV transmission, HPV natural history, cervical screening, treatment, and cancer survival that has been validated across several settings. The dynamic HPV transmission component of *Policy1-Cervix* incorporates HPV vaccination and was used to estimate HPV incidence by age under scenarios of disruption to vaccination coverage, compared to a no disruption scenario. The number of lifetime cervical cancer cases was subsequently estimated for the cohorts affected after explicitly modelling cervical screening (5-yearly HPV screening starting from age 25) (*Smith et al., 2021*) to take into account that cervical screening may prevent some cancers that would otherwise have been prevented by vaccination. In addition, HPV incidence estimated by the dynamic HPV transmission model was used in a separate incidence-based model to project the lifetime number of non-cervical HPV-related cancers in both sexes (i.e. anal, oropharyngeal, penile, vaginal, and vulvar), as previously described (*Kim et al., 2021*). Data included age- and sex-specific incidence of each cancer and proportion of cases attributable to vaccine-targeted HPV types (*Supplementary file 1A-C*). In a sensitivity analysis, we also used an analogous incidence-based approach to also estimate the impact on cervical cancer, rather than explicitly allowing screening to prevent some of the cervical cancers which would arise due to missed vaccination. Further information on the *Policy1-Cervix* model can be found at https://www.policy1.org/models/cervix/documentation.

### Scenarios
A baseline scenario of uninterrupted HPV vaccination (status quo) assumed a two-dose uptake of HPV9 at age 12 of 82.4% among females and 75.5% among males. Three disruption scenarios were modelled affecting females and males born in 2008 cohort who were aged 12 in 2020 (322,115

people). Scenario 1 assumed a fast catch-up where vaccination was delayed by one year (but no doses were entirely missed); scenario 2 assumed a slow catch-up where vaccination was delayed between 1 and 7 years with an equal proportion being caught up each year; and scenario 3 assumed there was no catch-up, and therefore any protection in the cohort would be due to herd effects only. A fourth scenario was also conducted which assumed individuals born in 2008 and in 2009 (644,230 people), missed HPV vaccination at age 12, with no catch-up.

For each scenario, we estimated the number of HPV-associated cancer cases (from age 12 to 84 years) prevented by vaccination compared to an unvaccinated comparator cohort, and the additional number of cases under each disruption scenario compared to the no disruption scenario.

## Additional information

### Competing interests

Megan A Smith: receives salary support via fellowship grants from the National Health and Medical Research Council (NHMRC) of Australia and Cancer Institute NSW and contracts paid to her institution (the Daffodil Centre) with the Commonwealth Department of Health (Australia) and National Screening Unit (New Zealand). Julia ML Brotherton: former employer ACPCC has received donated HPV tests and related items for validation and research purposes from Roche, Seegene, Abbott, Becton Dickinson, Cepheid and Copan. Karen Canfell: Co-PI of an investigator-initiated trial of cervical screening, "Compass", run by the Australian Centre for Prevention of Cervical Cancer (ACPCC), which is a government-funded not-for-profit charity. Compass receives infrastructure support from the Australian government and the ACPCC has received equipment and a funding contribution from Roche Molecular Diagnostics, USA. Co-PI on a major implementation program Elimination of Cervical Cancer in the Western Pacific which has received support from the Minderoo Foundation and the Frazer Family Foundation and equipment donations from Cepheid Inc. Receives contract funding from Commonwealth Department of Health, Australia to her institution for work to monitor the safety of the National Cervical Screening Program. Also receives support for a range of other Australian and international government projects including support from philanthropic organizations, WHO, and government agencies related to cervical cancer control. The other authors declare that no competing interests exist.

### Funding

| Funder | Grant reference number | Author |
| --- | --- | --- |
| National Health and Medical Research Council | APP1135172 | Karen Canfell |
| National Health and Medical Research Council | APP1159491 | Megan A Smith |
| Cancer Institute New South Wales | ECF181561 | Megan A Smith |

The funders had no role in study design, data collection, and interpretation, or the decision to submit the work for publication.

### Author contributions

Louiza S Velentzis, Conceptualization, Writing – original draft, Writing – review and editing; Megan A Smith, Conceptualization, Supervision, Visualization, Methodology, Writing – review and editing; James Killen, Formal analysis, Validation, Visualization, Writing – review and editing; Julia ML Brotherton, Rebecca Guy, Writing – review and editing; Karen Canfell, Conceptualization, Funding acquisition, Writing – review and editing

### Author ORCIDs

Louiza S Velentzis ![ORCID] https://orcid.org/0000-0002-9309-0492
Megan A Smith ![ORCID] https://orcid.org/0000-0002-0401-2653

Decision letter and Author response
Decision letter https://doi.org/10.7554/eLife.85720.sa1
Author response https://doi.org/10.7554/eLife.85720.sa2

## Additional files

### Supplementary files

• Supplementary file 1. Data used to inform the model. (A) Age-specific cancer rates for females in Australia (per 100,000), 2020 projections; (B) age-specific cancer rates for males in Australia (per 100,000), 2020 projections; (C) human papillomavirus HPV attributable fractions and HPV9 preventable proportions for the cancers modelled; (D) estimated number of cervical and total cancer cases in modelled scenarios for explicit screening (main analysis) vs incidence-based approach.

• MDAR checklist

### Data availability

Supporting information on data, data sources, model parametrization, calibration to epidemiologic data, and calibration approach in line with good modeling practice can be found partly in the supplementary materials for this article and partly online (https://www.policy1.org/models/cervix/documentation). The current manuscript is a computational study involving modelling rather than direct analysis of primary datasets and no new datasets have been created. Code for the *Policy1-Cervix* microsimulation model has been developed over decades, is proprietary property, and cannot be provided by the authors at this time. However, the Daffodil Centre is working to provide transparent and reproducible modelling code for forthcoming projects. Access to code is possible only through appropriately resourced supervision at the Daffodil Centre after submission and approval of a detailed proposal by the interested researcher. The corresponding authors should be contacted in the first instance.

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
