## [Editor Report]

The study presents important findings for public health authorities and policymakers to enable them to make evidence-based decisions when deciding on how to manage the effect of HPV vaccination disruptions. This study is particularly relevant in light of the efforts of the WHO to achieve global elimination of cervical cancers. The findings are convincing and the model used is appropriate.

---

## [Decision Letter]

**Decision letter after peer review:**

Thank you for submitting your article "The impact of COVID-19-related disruptions to HPV vaccination – a modelled analysis" for consideration by *eLife*. Your article has been reviewed by 2 peer reviewers, and the evaluation has been overseen by a Reviewing Editor and Diane Harper as the Senior Editor. The following individuals involved in the review of your submission have agreed to reveal their identity: Aura Timmen (Reviewer #1), Rebecca Luckett (Reviewer #2).

Essential revisions:

1. The abstract is difficult to follow. State more clearly what "natural history / vaccination / screening / HPV-related cancers" means in association with the model rather than putting it in parentheses. Similarly in the 3rd sentence, clarify what the "no disruption" scenario is as its own sentence, then explain what you are comparing it to. It may also help to have the 'n' of the background number of HPV-related cancers since n's are given later on for each scenario to give context to this finding. In the sentence starting with "additional". It is not clear what (males: 20%) and (females: 16%) mean after the two types of cancers from the abstract alone. These should either be explained or taken out. Finally, the difference between the fourth scenario and the "worst case scenario" is confusing.

2. In the first introductory paragraph, please explain the content in parentheses to improve clarity.

3. Methods: just as in the abstract, please clarify the fourth scenario and the difference in outcomes in the 2008 vs 2008-2009 affected cohorts.

4. Results section: Given there are multiple scenarios given, in the opening Results section, please add "in the baseline scenario of no interruption of HPV vaccination…" or something similar to set the stage very clearly.

5. There is space to expand on the results presented in Table 1, including an explanation of Affected cohorts 2008 vs Affected cohorts 2008-2009. It may also be useful to explain this analysis in the methods section.

*Reviewer #1 (Recommendations for the authors):*

Some considerations regarding how public debates about COVID-19 vaccination might have impacted the uptake of the HPV-vaccine might enrich the discussion.

*Reviewer #2 (Recommendations for the authors):*

– If 2020-2021 vaccination data is now available, could this be mentioned in the discussion (it is alluded to in the introduction).

– The abstract is difficult to follow. State more clearly what "natural history / vaccination / screening / HPV-related cancers" means in association with the model rather than putting it in parentheses. Similarly in the 3rd sentence, clarify what the "no disruption" scenario is as its own sentence, then explain what you are comparing it to. It may also help to have the 'n' of the background number of HPV-related cancers since n's are given later on for each scenario to give context to this finding. In the sentence starting with "additional". It is not clear what (males: 20%) and (females: 16%) mean after the two types of cancers from the abstract alone. These should either be explained or taken out. Finally, the difference between the fourth scenario and the "worst case scenario" is confusing.

– In the first introductory paragraph, I would recommend explaining the content in parentheses to improve clarity.

– Methods: just the same comment as in the abstract to clarify the fourth scenario and the difference in outcomes in the 2008 vs 2008-2009 affected cohorts.

– Results section: Given there are multiple scenarios given, in the opening Results section, I would suggest adding "in the baseline scenario of no interruption of HPV vaccination…" or something similar to set the stage very clearly.

– There is space to expand on the results presented in Table 1, including an explanation of Affected cohorts 2008 vs Affected cohorts 2008-2009. It may also be useful to explain this analysis in the methods section.

– Given that Australia is a best-case scenario and other countries have not had the same success in HPV vaccination coverage, in the discussion would it be possible to give a comparison of how these three scenarios would look different in a population with school-based vaccination but lower coverage volume such that readers could understand how much of the success / failures of each of the three catch-up scenarios? It would be particularly helpful for readers who are not familiar with the modeling tool used in this analysis.

---

## [Author Response]

Essential revisions:1. The abstract is difficult to follow. State more clearly what "natural history / vaccination / screening / HPV-related cancers" means in association with the model rather than putting it in parentheses.

We have removed the parenthesis and provided a clearer explanation of the model used. The relevant sentence has now been changed as shown below:

“We used ‘Policy1-Cervix’, a dynamic model simulating HPV natural history, sexual behaviour, HPV transmission, vaccination, cervical screening, and diagnosis of HPV-related cancers, to estimate the impact on HPV-related cancers from disruptions to HPV vaccination in a high-income setting.”

Similarly in the 3rd sentence, clarify what the "no disruption" scenario is as its own sentence, then explain what you are comparing it to.

The ‘no disruption’ scenario has now been written in a single sentence as below:

“A baseline scenario of no disruption to HPV vaccination was modelled, which assumed uptake of the nonavalent vaccine at age 12 by 82.4% among females and 75.5% among males, as is the coverage in Australia.”

It may also help to have the 'n' of the background number of HPV-related cancers since n's are given later on for each scenario to give context to this finding.

The background number of cases for the baseline ‘no disruption’ scenario has now been added as shown below:

“Compared to 1,532 HPV-related cancer cases estimated for the baseline no disruption scenario, we found a 1-year delay could result in ≤0.3% more HPV-related cancers (n=4) but the increase would be greater if catch-up was slower (5%; n=70), and especially if there was no catch-up (49%; n=750).”

In the sentence starting with "additional". It is not clear what (males: 20%) and (females: 16%) mean after the two types of cancers from the abstract alone. These should either be explained or taken out.

The sentence has been amended as shown below for additional clarity:

“Additional cancers for a single missed cohort were most commonly cervical (23%) and anal cancers (16%) in females and oropharyngeal cancers in males (20%).”

Finally, the difference between the fourth scenario and the "worst case scenario" is confusing.

We have amended the sentence describing the fourth scenario modelled as shown below, which was also the worst-case scenario, as individuals born in 2008 and in 2009 would not receive HPV vaccination.

“A fourth scenario assumed no catch-up HPV vaccination for two birth cohorts, i.e. all individuals born in 2008 and in 2009 missed vaccination (worst case scenario).

2. In the first introductory paragraph, please explain the content in parentheses to improve clarity.

We have now amended the relevant sentence as indicated below:

“Additionally, catch-up HPV vaccination was offered to females aged 14 to 26 years in 2007-2009 and to males aged 14-15 in 2013-2014.”

3. Methods: just as in the abstract, please clarify the fourth scenario and the difference in outcomes in the 2008 vs 2008-2009 affected cohorts.

We have amended the sentence explaining the fourth scenario more clearly, as shown below:

“A fourth scenario was also conducted which assumed individuals born in 2008 and in 2009 (644,230 people), missed HPV vaccination at age 12, with no catch-up.”

4. Results section: Given there are multiple scenarios given, in the opening Results section, please add "in the baseline scenario of no interruption of HPV vaccination…" or something similar to set the stage very clearly.

We have amended the relevant sentence as indicated:

“In the baseline scenario of no interruption to HPV vaccination, we estimate 2,583 HPV-related cancer cases would be prevented in the 2008 cohort due to HPV vaccination, equating to 63% of all HPV-related cases (Table 1).”

5. There is space to expand on the results presented in Table 1, including an explanation of Affected cohorts 2008 vs Affected cohorts 2008-2009. It may also be useful to explain this analysis in the methods section.

Table 1 has been updated to refer to “Outcomes in 2008 birth cohort” rather than “Affected cohort (2008)” and footnotes have been added to further clarify the affected cohorts.

“* Includes outcomes specifically for the cohort consisting of females and males born in 2008 (any effects on the 2009 cohort are not included).

** Includes outcomes specifically for the cohort consisting of females and males born in either 2008 or 2009.”

In the methods section, reference had previously been included regarding the cohorts affected by the scenarios modelled. However, further clarification in wording has been provided as seen below:

“Three disruption scenarios were modelled affecting females and males born in 2008 who were aged 12 in 2020 (322,115 people)…. A fourth scenario was also conducted which assumed individuals born in 2008 and in 2009 (644,230 people), missed HPV vaccination at age 12, with no catch-up.”

Reviewer #1 (Recommendations for the authors):Some considerations regarding how public debates about COVID-19 vaccination might have impacted the uptake of the HPV-vaccine might enrich the discussion.

In the past, vaccine hesitancy impacted uptake of HPV vaccination in certain settings like Japan, Denmark and Ireland (Bruni et al. Lancet Glob Health. 2016 Jul;4(7):e453-63). However, in Australia, coverage rates have been relatively high and improved since the National HPV Vaccination Program Register was established. In the Australian context, there isn’t strong evidence that the COVID-19 vaccine debate and hesitancy had any substantive impact on HPV vaccine uptake. We agree with the reviewer that a complex interaction between COVID-19 vaccine hesitancy and HPV vaccine hesitancy may have played out and impacted coverage in other settings. We have briefly referred to data from Europe indicating that there was a loss in confidence in the HPV vaccine between 2020 and 2022 in many countries.

“Globally, the timeframe to reach elimination is likely to be impacted by pandemic-induced delays in HPV vaccination delivery and also by global shortages of the HPV vaccine estimated to last until 2024 thus delaying the introduction of vaccination in low- and middle-income countries.1415 In countries predicted to achieve cervical cancer elimination in the relatively short term, such as Australia, short delays of one year are not predicted to delay elimination.^8^ There are signs, however, of a decrease in vaccine confidence in Europe around the time of the COVID-19 pandemic, with lower confidence for the HPV vaccine reported in 17 countries in 2022 compared to 2020.”

Reviewer #2 (Recommendations for the authors):– If 2020-2021 vaccination data is now available, could this be mentioned in the discussion (it is alluded to in the introduction).

Final coverage data is reported as those who are vaccinated by age 15, and this is not yet available for the most affected cohorts (aged 12-13 in 2020 and 2021; born approximately 2007-2009; turning 15 in 2022-2024). The latest coverage available is for those turning 15 in 2021, who were mostly offered vaccination at school in 2019 or earlier. To clarify this, we have updated the discussion as seen below:

“In Australia, initial national figures show for adolescents aged 11 to <15 years who received their first dose in 2020, 74.7% of girls and 72.6% of boys, received their second dose in the same calendar year compared to 86.2% and 84.3% for girls and boys, respectively, in 2019.^2^ The difference in numbers suggest vaccination delays, possibly due to COVID-19 effects including school closures during lockdowns, and decreased school attendance due to infections. In line with WHO recommendations, final coverage data is reported as those who are vaccinated by age 15, and is not yet available for the most affected cohorts (aged 12-13 in 2020 and 2021; born approximately 2007-2009; turning 15 in 2022-2024).^10^”

– The abstract is difficult to follow. State more clearly what "natural history / vaccination / screening / HPV-related cancers" means in association with the model rather than putting it in parentheses.

This has been addressed above (editor’s essential revision 1), however, the response is also provided below for ease of access We have removed the parenthesis and provided a clearer explanation of the model used. The relevant sentence has now been changed to:

“We used ‘Policy1-Cervix’, a dynamic model simulating HPV natural history, sexual behaviour, HPV transmission, vaccination, cervical screening, and diagnosis of HPV-related cancers, to estimate the impact on HPV-related cancers from disruptions to HPV vaccination in a high-income setting.”

Similarly in the 3rd sentence, clarify what the "no disruption" scenario is as its own sentence, then explain what you are comparing it to.

The ‘no disruption’ scenario has now been written in a single sentence as below:

“A baseline scenario of no disruption to HPV vaccination was modelled, which assumed uptake of the nonavalent vaccine at age 12 by 82.4% among females and 75.5% among males, as is the coverage in Australia.”

It may also help to have the 'n' of the background number of HPV-related cancers since n's are given later on for each scenario to give context to this finding.

The background number of cases for the baseline ‘no disruption’ scenario has now been added as shown below:

“Compared to 1,532 HPV-related cancer cases estimated for the baseline no disruption scenario, we found a 1-year delay could result in ≤0.3% more HPV-related cancers (n=4) but the increase would be greater if catch-up was slower (5%; n=70), and especially if there was no catch-up (49%; n=750).”

In the sentence starting with "additional". It is not clear what (males: 20%) and (females: 16%) mean after the two types of cancers from the abstract alone. These should either be explained or taken out.

The sentence has been amended as shown below in bold for additional clarity:

“Additional cancers for a single missed cohort were most commonly cervical (23%) and anal cancers (16%) in females and oropharyngeal cancers in males (20%).”

Finally, the difference between the fourth scenario and the "worst case scenario" is confusing.

We have amended the sentence describing the fourth scenario modelled as shown below, which was also the worst-case scenario, as individuals born in 2008 and in 2009 would not receive HPV vaccination.

“A fourth scenario assumed no catch-up HPV vaccination for two birth cohorts, i.e. all individuals born in 2008 and in 2009 missed vaccination (worst case scenario).”

– In the first introductory paragraph, I would recommend explaining the content in parentheses to improve clarity.

We have now amended the relevant sentence as indicated below:

“Additionally, catch-up HPV vaccination was offered to females aged 14 to 26 years in 2007-2009 and to males aged 14-15 in 2013-2014.”

– Methods: just the same comment as in the abstract to clarify the fourth scenario and the difference in outcomes in the 2008 vs 2008-2009 affected cohorts.

We have amended the sentence explaining the fourth scenario more clearly, as shown below:

“A fourth scenario was also conducted which assumed individuals born in 2008 and in 2009 (644,230 people), missed HPV vaccination at age 12, with no catch-up.”

– Results section: Given there are multiple scenarios given, in the opening Results section, I would suggest adding "in the baseline scenario of no interruption of HPV vaccination…" or something similar to set the stage very clearly.

We have amended the relevant sentence as indicated:

“In the baseline scenario of no interruption to HPV vaccination, we estimate 2,583 HPV-related cancer cases would be prevented in the 2008 cohort due to HPV vaccination, equating to 63% of all HPV-related cases (Table 1).”

– There is space to expand on the results presented in Table 1, including an explanation of Affected cohorts 2008 vs Affected cohorts 2008-2009. It may also be useful to explain this analysis in the methods section.

Please refer to response provided to editor on the same question (comment 5).

– Given that Australia is a best-case scenario and other countries have not had the same success in HPV vaccination coverage, in the discussion would it be possible to give a comparison of how these three scenarios would look different in a population with school-based vaccination but lower coverage volume such that readers could understand how much of the success / failures of each of the three catch-up scenarios? It would be particularly helpful for readers who are not familiar with the modeling tool used in this analysis.

We have added some commentary in the discussion in response to the reviewer’s comment. In future, further similar work in countries with lower base coverage would be informative.

“Australia is a relatively high HPV vaccination coverage setting. Outcomes may be less favourable in a lower coverage setting, as there would be less protection from herd effects; however, the impact of disruptions might also be smaller in a setting with lower coverage, since a lower coverage program would be less effective. Nevertheless, the finding that if catch-up is performed expeditiously then it mitigates much of the effect from vaccination delays, is likely to hold in other settings. In a previous study (Simms et al., Lancet Public Health. 2020 Apr;5(4): e223-e234) modelling the health impacts of HPV vaccination hesitancy in Japan from 2013 to 2019 and the potential effects of restoring coverage to 70% with catch-up vaccination in 2020 is informative as it demonstrates that multi-age HPV catch-up vaccination, after catastrophic falls in coverage in Japan, would be effective in mitigating the effects.”